# Anti-Atopic Effect of Acorn Shell Extract on Atopic Dermatitis-Like Lesions in Mice and Its Active Phytochemicals

**DOI:** 10.3390/biom10010057

**Published:** 2019-12-29

**Authors:** Sullim Lee, Hyun Jegal, Sim-Kyu Bong, Kyeong-No Yoon, No-June Park, Myoung-Sook Shin, Min Hye Yang, Yong Kee Kim, Su-Nam Kim

**Affiliations:** 1Department of Life Science, College of Bio-Nano Technology, Gachon University, Seongnam 13120, Korea; sullimlee@gachon.ac.kr; 2Natural Products Research Institute, Korea Institute of Science and Technology, Gangneung 25451, Korea; wprkfgus3@gmail.com (H.J.); 115044@kist.re.kr (S.-K.B.); kyeongnoyoon@naver.com (K.-N.Y.); parknojune@kist.re.kr (N.-J.P.); 3College of Korean Medicine, Gachon University, Seongnam 13120, Korea; ms.shin@gachon.ac.kr; 4College of Pharmacy, Pusan National University, Busan 46241, Korea; mhyang@pusan.ac.kr; 5College of Pharmacy, Sookmyung Women′s University, Seoul 04610, Korea

**Keywords:** acorn shell, atopic dermatitis, IL-4, gallic acid, ellagic acid

## Abstract

To investigate the potential effects of acorn shells on atopic dermatitis (AD), we utilized oxazolone (OX)- or 2,4-dinitrochlorobenzene (DNCB)-induced AD-like lesion mouse models. Our research demonstrates that Acorn shell extract (ASE) improved the progression of AD-like lesions, including swelling, which were induced by oxazolone on Balb/c mouse ears. Additionally, ASE significantly decreased the ear thickness (OX: 0.42 ± 0.01 mm, OX-ASE: 0.32 ± 0.02 mm) and epidermal thickness (OX: 75.3 ± 32.6 µm, OX-ASE: 46.1 ± 13.4 µm). The continuous DNCB-induced AD mouse model in SKH-1 hairless mice demonstrated that ASE improved AD-like symptoms, including the recovery of skin barrier dysfunction, Immunoglobulin E hyperproduction (DNCB: 340.1 ± 66.8 ng/mL, DNCB-ASE: 234.8 ± 32.9 ng/mL) and an increase in epidermal thickness (DNCB: 96.4 ± 21.9 µm, DNCB-ASE: 52.4 ± 16.3 µm). In addition, we found that ASE suppressed the levels of AD-involved cytokines, such as Tumor Necrosis Factor α, IL-1β, IL-25 and IL-33 in both animal models. Furthermore, gallic acid and ellagic acid isolated from ASE suppressed β-hexosaminidase release and IL-4 expression in RBL-2H3 cells. The acorn shell and its active phytochemicals have potential as a new remedy for the improvement of atopic dermatitis and other inflammatory diseases.

## 1. Introduction 

Atopic dermatitis (AD) is a chronic inflammatory skin disease with symptoms including lesions from severe pruritus, epidermal hyperplasia, edema, erythema and erythematous plaque [1,2]. The cause of AD has not yet been identified, but genetic factors, immune function imbalance and environmental factors are known to be involved [3,4]. AD is estimated to occur in approximately 10%–20% of the total population, but there is no clear curative treatment. Immunologically, activation of T lymphocytes, abnormalities in the cytokine system, decreases in cell mediated immunity and increases in IgE, have been reported to play an important role in the pathology of AD [5]. 

Atopic skin inflammation is regulated by diverse inflammatory cytokines. According to previous reports, the cytokines Interleukin 4 (IL-4), IL-5 and IL-13 are hyperproduced in AD patients compared with healthy patients, and IL-4 and Interferon-γ are clearly increased in acute AD [6,7]. Currently, topical ointments and oral medications are used as AD therapies to suppress inflammation and itching [8]. Generally, steroids, antihistamines, immunosuppressants and immunosuppressive calcineurin inhibitors are frequently used as therapeutics [8,9,10]. However, long term use of these agents causes side effects such as extreme atrophy of the skin, adrenal suppression and sensitivity to infection [9,10,11,12,13]. For these reasons, research is focused on identifying remedies for AD with few adverse effects. Therefore, we are conducting research focused on the development of new remedies originating from natural resources with low adverse drug reactions.

Acorns from up to 600 *Quercus* species are widely distributed worldwide and play an important role in food production and livestock husbandry [14,15,16]. Acorns have also been originally used in traditional Chinese medicine and folk medicine for diarrhea and diverse inflammatory disorders [17,18]. In Iran, its shells have been used as a folk remedy for wound healing [19]. According to previous reports, various acorns have also been reported to have a variety of bioactivities, including free radical scavenging [20], anti-bacterial activity [21], anti-inflammatory activity [17] and antifungal activity [22]. However, the major components with these bioactivities are mainly secondary metabolites, which are found in the shell more than in the fruit [23]. Especially, the acorn shell has been reported to have angiogenesis activity in vitro [19]. Angiogenesis plays an important role in tissue regeneration for atopic diseases and the onset of allergic inflammation [24]. Thus, the aim of this research is to propose acorn shells as a possible remedy for atopic dermatitis, since they are often abandoned and have anti-inflammatory effects.

## 2. Materials and Methods 

### 2.1. Chemicals

4-ethoxymethylene-2-phenyl-2-oxazolin-5-one (oxazolone) 2,4-dinitrochlorobenzene (DNCB) and other organic solvents were obtained from Sigma Aldrich (St. Louis, MO, USA). Dulbecco′s Modified Eagle′s Media (DMEM) and fetal bovine serum (FBS) were purchased from HyClone (Logan, UT, USA) and Median Life Science (Houston, TX, USA), respectively.

Penicillin–streptomycin solution and Fast SYBR^®^ Green Master Mix were purchased from Life Technologies (Waltham, MA, USA). RNeasy mini kit, and ImProm-II Reverse Transcription System kit were purchased from Qiagen (Valencia, CA, USA) and Promega (Madison, WI, USA), respectively.

### 2.2. Sample Collection and Extraction

The acorns from *Quercus mongolica* were collected in September 2015 from the Wonju, Gangwon-do area in Korea. The voucher specimen (SN20151001) has been deposited at the Natural Products Research Institute of the Korea Institute of Science and Technology (KIST). The dried acorn shell (46.5 g) was extracted thrice with 95% ethanol (0.5 L) and evaporated under vacuum to yield the acorn shell extract (ASE, 3.6 g) for further studies.

### 2.3. Mice

Six-week old female Balb/c and SKH-1 hairless mice, were obtained from the animal facility (Orient Bio Inc., Seongnam, Korea) and maintained under constant temperature and humidity (23 ± 2 °C and 55% ± 5%) in the animal laboratory. All animal experiments were approved by the Institutional Animal Care and Use Committee of the KIST (KIST-2016-011) and performed based on the Guide for the Care and Use of Laboratory Animals of the National Institutes of Health (NIH publication No. 85-23, revised on 26 January 1996). 

### 2.4. Induction of AD-Like Skin Lesions in Balb/c Mice by Oxazolone and ASE Application

As described previously, AD was induced onto the ears of Balb/c mice with oxazolone (OX) dissolved in a mixture of acetone and olive oil (4:1) [25]. Briefly, over 7 days, 1% oxazolone was applied to the ears daily in both the OX (oxazolone)-treated group and the oxazolone and 1% ASE (OX-ASE)-treated group. Starting on day 8, 0.1% oxazolone (20 μL) was applied to the ears every other day for 3 weeks. During that same period, 1% ASE (20 μL) was applied daily to the ears of mice in the OX-ASE-treated group 4 h after oxazolone application. Meanwhile, distilled water was applied to mice in the control (CON) group instead of oxazolone. On the final day of the experiment, AD-like skin lesions, including erythema and ear swelling, were measured. Ear thickness was measured using a micrometer (Mitutoyo Corp., Kawasaki, Japan), and applied to the ear edge immediately adjacent to the cartilage bulge and recorded. After sacrificing the mice, ear skins were collected from the mice for further studies. 

### 2.5. Induction of AD-Like Skin Lesions in SKH-1 Hairless Mice by DNCB and ASE Application

As in our previously published report, AD was induced by DNCB dissolved in acetone in the dorsal skin of SKH-1 hairless mice [25]. Briefly, over 7 days, 1% DNCB was applied to the dorsal skin of SKH-1 hairless mice once a day in both the DNCB-treated group and the DNCB and 1% ASE (DNCB-ASE)-treated group. Starting on day 8, 0.1% DNCB (100 μL) was applied every other day for an additional 2 weeks. During that same period, 1% ASE (100 μL) was applied to mice in the DNCB–ASE group 4 h after DNCB application. Meanwhile, distilled water was applied to the CON group. On the last day, erythema, transepidermal water loss (TEWL) and skin hydration were evaluated to determine the degree of dermatitis in SKH-1 hairless mice. TEWL and hydration were measured using the Tewameter^®TM^ 210 (Courage and Khazaka, Cologne, Germany) and SKIN-O-MAT (Cosmomed, Ruhr, Germany) under 23 ± 2 °C and 55% ± 5% humidity conditions. On the final day, mice were sacrificed and blood and skin tissue were collected for further studies. 

### 2.6. Histopathological Evaluation of Skin Lesions

To evaluate histopathological characteristics, ears from Balb/c mice and dorsal skin from SKH-1 hairless mice were fixed with formalin and covered with paraffin wax. Sections (10 µm) were cut using a microtome, then dried and stained with hematoxylin and eosin (H&E) and toluidine blue, respectively. Histopathological appearances (200× magnification) were observed using an Olympus CX31/BX51microscope (Olympus Optical Co., Tokyo, Japan) and TE-2000U camera (Nikon Instruments Inc., Melville, NY, USA). Epidermis thicknesses were measured by determining the length from the stratum corneum to the stratum basale using a microscope equipped with a ruler and the LAS v4.8 (Leica Microsystem, Herbrugg, Switzerland) program. 

### 2.7. Determination of Serum IgE and IL-4 Concentration

Blood samples were taken from the abdominal aorta of mice when they were sacrificed. The serum was immediately collected from the blood by centrifugation and stored at −80 °C for additional experiments. Serum IgE and IL-4 concentrations were determined by mouse IgE and IL-4 ELISA kits (eBioscience, San Diego, CA, USA), respectively, according to the manufacturer′s protocol.

### 2.8. HPLC Analysis of ASE

High-performance liquid chromatography (HPLC) analysis was performed by an Agilent 1200 Series HPLC/MS system (Agilent Technologies, Palo Alto, CA, USA). ASEs were dissolved in 50% methanol and filtered through a 0.45 µm polytetrafluoroethylene (PTFE) membrane filter. The column used for separation was a YMC-Triart C_18_ column (5 µm, 4.6 × 150 mm), and the flow rate of the mobile phase was 1.0 mL/min under a gradient of 0.05% trifluoroacetic acid (TFA) in methanol and 0.05% TFA in water.

### 2.9. Cells and Cell Culture

The RBL-2H3 cell line is a rat bassophilic leukemia cell line that was obtained from the American Type Culture Collection (ATCC, #CRL-2256, Bethesda, MD, USA). Cells were cultured in DMEM containing 10% FBS, and 1% penicillin-streptomycin in a CO_2_ incubator at 37 °C.

### 2.10. Determination of Gene Expression

Cells and tissues were collected to synthesize cDNA, and mRNA expression was measured using qPCR (quantitative real-time PCR). Total RNA isolation and cDNA synthesis were performed using the RNeasy mini kit and ImProm-II Reverse Transcription System, respectively. qPCR was performed using the 7500 Fast Real-Time PCR system (Applied Biosystems, Singapore) and Fast SYBR^®^ Green Master Mix. All reaction experiments were independently repeated three times, and the data were analyzed using the comparison cycle threshold (Ct) method [26]. PCR primers for IL-4 (NM_021283) were 5′-ACC TTG CTG TCA CCC TGT TC-3′ (forward), 5′-TTG TGA GCG TGG ACTCAT TC-3′ (reverse); TNFα (NM_013693) were 5′-AGC CCC CAG TCT GTA TCC TT-3′ (forward) and 5′-CTC CCT TTG CAG AAC TCA GG-3′ (reverse); IL-1β (NM_008361) were 5′-CAA CCA ACA AGT GAT ATT CTC CAT G-3′ (forward) and 5′-GAT CCA CAC TCT CCA GCT GCA-3′ (reverse); IL-25 (NM_080729) were 5′-CAG CAA AGA GCA AGA ACC-3′ (forward) and 5′-CCC TGT CCA ACT CAT AGC-3′ (reverse); IL-33 (NM_133775) were 5′-CAA TCA GGC GAC GGT GTG GAT GG-3′ (forward) and 5′-TCC GGA GGC GAG ACG TCA CC -3′(reverse); for β-actin (NM_007393) were 5′-TCA TCA CCA TCG GCA ACG-3′ (forward), 5′-TTC CT GAT GTC CAC GTC GC-3′ (reverse) [27]. The mRNA expression was normalized with β-actin and calculated based on the ratio to 100% of the phorbol 12-myristate 13-acetate (PMA)/ionomycin-treated group, the oxazolone-treated group or DNCB-treated group.

### 2.11. Determination of β-Hexosaminidase Release

Degranulation of RBL-2H3 cells was measured by determining the inhibitory effect of β-hexosaminidase secretion, which is a biomarker of degranulation [28]. The secretion of β-hexosaminidase was measured in order to investigate the inhibitory effect on degranulation, an indicator of an immediate allergic response. RBL-2H3 cells were seeded into a 24-well plate and cultured overnight. Over 4 h, these cells were sensitized with anti-dinitrophenyl immunoglobulin E (anti-DNP IgE) and then treated with ASE and its fractions (10 and 30 µg /mL). After incubating for 1 h, cells were stimulated by 100 ng/mL of DNP-BSA antigen (100 ng/mL) at 37 °C for 30 min. Since the β-hexosaminidase released from the cells was present in the cell culture medium, the supernatant culture medium was transferred to a new 96-well plate for the next step. The substrate (10 mm poly-*N*-acetyl glucosamine (p-NAG) in 0.1 M sodium citrate buffer, pH 4.5) was added to the supernatants, mixed and incubated at 37 °C for 2 h. The reaction was terminated by adding a stop solution (0.1 M sodium carbonate buffer, pH 10.0). Absorbance measurements were performed using an infinite M1000 (TECAN, Salzburg, Austria) microplate reader at 405 nm. The amount of β-hexosaminidase released from mice in each group was calculated based on a ratio to 100% of the untreated group.

### 2.12. Determination of β-Hexosaminidase Release

ASE fractionation was carried out using a recycling preparative HPLC system. The JAI LC-9014/L-6050 system (JAI, Tokyo, Japan) was used for recycling preparation, and the UV-3702 system was used as the UV detector. These experiments were performed with peak reproducibility over 95%. The purity and molecular weight of the active phytochemicals were measured with an HPLC/MS. Structures were identified by ^1^H- and ^13^C-NMR analysis using a Bruker AVANCE 500 nuclear magnetic resonance (NMR) spectrometer (Bruker, Rheinstetten, Germany).

### 2.13. Statistical Analysis

All quantitative data for this study were obtained through at least two independent experiments, and are denoted as mean ± standard deviation (SD). Statistical analyses were carried out by a one-way analysis of variance (ANOVA) and a Tukey′s multiple comparisons post hoc analysis.

## 3. Results and Discussion

### 3.1. Effects of ASE Application on Oxazolone-Induced AD-Like Skin Lesions Mice

Atopic dermatitis causes the excessive production of serum IgE and T-helper cell type-2 (Th2) expansion [29]. Generally, patients with AD have an increased production of Th2-mediated inflammatory cytokines including IL-4, IL-5 and IL-13 [30,31,32]. Among these cytokines, IL-4 is known to play an important role for IgE hyperproduction which occurs at the onset of atopic dermatitis [33,34]. Therefore, IL-4 inhibitors could be promising AD therapeutic agents. During our preliminary search for IL-4 inhibitors, we confirmed that ASE inhibits the mRNA expression of IL-4 in RBL-2H3 cells. Additionally, acorn shells have been reported to have angiogenesis activity, which plays an important role in the pathophysiology of tissue remodeling in atopic dermatitis [19]. These preliminary experiments and reports predicted that the acorn shell was effective for the improvement of atopic dermatitis. 

Haptens, including oxazolone and DNCB, are small molecules that can effectively penetrate the skin barrier and epidermis of undamaged mouse skin. Proper exposure raises an acquired immune response, which causes a contact hypersensitivity reaction, such as allergic contact dermatitis. Repeated induction occurs with the type 1 and type 2 allergic contact dermatitis, creating a state that is like atopic dermatitis [35,36]. However, the pathogenesis of allergic contact dermatitis and atopic dermatitis are not exactly the same. Therefore, it is difficult for these mouse models to clearly reappear atopic dermatitis. Although the mouse models are not a complete imitation of the overall clinical symptoms, it does provide a basic core of phenotypic expression [37,38]. Thus, using the mouse models, we tested the effects of acorn shell extract (ASE) that induces AD-like skin lesions with oxazolone and DNCB.

First, to investigate the effect of ASE topical application on atopic dermatitis, we performed an experiment with oxazolone-induced AD-like skin lesions in ears of Balb/c mice. In this AD mouse model, stimulation induced by oxazolone has previously shown typical allergic responses including increases in ear and epidermal thickness, swelling, dryness and erythema [39]. As a result of the phenotypic observation, ASE application improved AD-like skin lesions, such as swelling, dryness and erythema (Figure 1A). The oxazolone-treated group (OX) demonstrated increased ear and epidermal thickness, both of which were suppressed in the ASE-treated group (OX-ASE). The application of ASE showed a significant decrease in ear thickness (CON: 0.20 ± 0.02 mm, OX: 0.42 ± 0.01 mm, OX-ASE: 0.32 ± 0.02 mm, Figure 1B,C). In addition, the epidermal thickness of mice in the ASE-treated group (OX-ASE: 46.1 ± 13.4 µm) decreased compared to the oxazolone-treated group (OX: 75.3 ± 32.6 µm, Figure 1D). The atopic dermatitis response induces a variety of responses in the immune system, and this reaction can primarily affect the weight of the immune system. In AD skin lesions, numerous pro-inflammatory cytokines are secreted by allergens, leading to an allergic immune response. Activated keratinocytes secrete inflammatory cytokines such as TNFα, which increase the infiltration of eosinophils and lymphocytes, and they secrete a large number of cytokines, several of which are Th2 cytokines [40]. Therefore, mRNA gene expression of the pro-inflammatory cytokines such as TNFα, IL-1β, IL-33 and IL-25 and IL-4 were measured in tissues. The mRNA gene expressions of TNFα, IL-1β, IL-33, IL-25 and IL-4 were increased in the OX group (TNFα: 1.81 ± 0.07, IL-1β: 4.12 ± 0.54, IL-33: 4.44 ± 0.65, IL-4: 2.80 ± 0.14 fold) compared to the CON group, whereas ASE significantly decreased gene expressions (TNFα: 1.17 ± 0.23, IL-1β: 1.64 ± 0.63, IL-33: 1.40 ± 0.10, IL-4: 1.23 ± 0.13 fold) (Figure 1E). These results showed that topical application of ASE suppressed allergic immune responses. 

The oxazolone-induced AD mouse model experiment demonstrated that ASE improves AD-like lesions and allergic immune response in Balb/c mice ears. In this experiment, epidermal thickness and infiltration of mast cells increased with oxazolone, whereas ASE significantly reversed these features. In addition, the expression levels of inflammatory cytokines increased with oxazolone, whereas ASE suppressed. These results demonstrated that the topical application of ASE improved AD-like lesions on the ears and allergic immune response of oxazolone-induced Balb/c mice.

### 3.2. Effects of ASE Application on DNCB-Induced AD-Like Skin Lesions Mice

To investigate the more specific application effects of ASE on atopic dermatitis, we performed an experiment with DNCB-induced AD-like skin lesions on the dorsal skin of SKH-1 hairless mice. Stimulation induced by DNCB causes anaphylaxis and the onset of AD-like skin lesions including increases in epidermal thickness, swelling, dryness and erythema in an allergy-induced mouse model [41]. In our study, the DNCB-treated group (DNCB) demonstrated AD-like skin lesions such as swelling, dryness and erythema on the dorsal skin of SKH-1 hairless mice, whereas the ASE-treated group (DNCB-ASE) showed that AD-like skin lesions were improved (Figure 2A). In humans, mast cells are associated with the epithelium of the lungs around the alveoli, nasal mucosa, conjunctiva and intestinal mucosa, and serve as the first line of defense against parasitic infestations. Mast cells interact with allergens, such as ticks, and lead to allergies. The antibody is a protein called an immunoglobulin, and there are five types: IgE, IgA, IgG, IgM and IgD. The majority of allergies are caused by IgE. When allergens bind to IgE antibodies, chemicals such as histamine and leukotriene are released from mast cells. When an allergic reaction occurs, cells release inflammatory responses due to the release and inflow of chemical mediators. In this case, various types of cytokines, including interleukins, interferons and tumor necrosis factors (TNFs), directly or indirectly participate. Among them, IL-4 is a regulator of IgE, a mediator of type I allergic reactions, and mast cell-mediated immune responses; IL-4 is known to be a major contributor to the development of allergies [42]. Treatment with DNCB causes augmentation of serum IgE and IL-4 hyperproduction in rodent allergy models [25,43]. The DNCB group demonstrated increased serum IgE and IL-4 production, but these were suppressed in the DNCB-ASE group. Serum IgE production was increased in the DNCB group (340.1 ± 66.8 µg/mL) compared to the CON group (101.1 ± 60.36 µg/mL), whereas ASE decreased serum IgE concentrations to 234.8 ± 32.9 µg/mL (Figure 2B). Serum IL-4 production was increased in the DNCB group (44.1± 8.8 µg/mL) compared with the CON group (8.1 ± 11.0 µg/mL), but ASE decreased serum IL-4 concentrations to 26.5 ± 16.3 µg/mL (Figure 2C). These results indicate that the application of ASE could suppress the excessive production of serum IgE and IL-4. Thus, ASE is expected to down-regulate immune responses through the inhibition of IgE and Th2 cell expansion. 

An increase in epidermal moisture loss (TEWL) is usually found before the development of clinical AD. Previous studies have shown that sensitization with DNCB is related to skin barrier dysfunction, such as increases in TEWL and decreases in skin hydration [43,44]. In this study, the DNCB group also demonstrated increases in TEWL and decreases in skin hydration, whereas the DNCB-ASE group reversed these indicators. After 21 days of treatment, the TEWL level was increased in the DNCB group (88.9 ± 6.4 J [g/(m^2^ h)]), compared to the control group (29.5 ± 1.0 J [g/(m^2^ h)]), whereas ASE application increased the hydration level to 48.0 ± 15.4 J [g/(m^2^ h)] (Figure 2D). In conjunction with this result, the hydration level was decreased in the DNCB group (27.1 ± 3.0%) compared with the CON group (54.7± 5.5%), whereas ASE application increased the hydration level to 43.1 ± 8.6% (Figure 2E). In the development of clinical AD, impaired skin barrier leads to an increase of transepidermal water loss (TEWL) and decreases of skin hydration, and then it could be the reason for aggression and triggering inflammation, and be correlated with high IgE levels, pruritus, and asthma [45]. Because the impaired skin barrier aggravates AD symptoms, these results indicate that the application of ASE could improve the AD symptom through reverse skin barrier dysfunction.

The DNCB-induced AD mouse model experiment demonstrated that ASE improves AD-like lesions and allergic immune response in SKH-1 hairless mice. In this experiment, serum IgE and IL-4 production, epidermal thickness and the infiltration of mast cells increased with DNCB, whereas ASE significantly reversed these changes. In addition, sensitization with DNCB caused skin barrier dysfunction, including increases in TEWL and decreases in hydration, whereas ASE recovered these skin barrier dysfunctions. ASE was found to have anti-inflammatory effects on atopic dermatitis and be effective in the improvement of symptoms. ASE was especially effective in suppressing the production of IgE caused by the allergic reaction. Therefore, ASEs could be widely used for the treatment of atopic dermatitis. 

This is consistent with the existing immunological mechanism of action, thus, the anti-allergic effects through the immunoregulatory actions of the present ASE is demonstrated. 

In histological experiments, dermal infiltration with mast cells and eosinophils has been observed before the clinical AD skin lesion appeared [46]. Eosinophils act as immune modulators by secreting various chemokines and cytokines that recruit more eosinophils to the inflammation area [47]. Our histological analysis also demonstrated that the application of DNCB caused AD-like symptoms including inflammatory cell infiltration and edema lesions, whereas ASE reversed these symptoms (Figure 3A,B). 

In the ASE-treated group, the epidermal thickness (CON: 23.0 ± 4.3 µM, DNCB: 96.4 ± 21.9 µM, DNCB-ASE: 52.4 ± 16.3 µM, Figure 3C) and the number of infiltrating lymphocytes (CON: 70.0 ± 21.2, DNCB: 153.9 ± 75.8, DNCB-ASE: 81.8 ± 40.7, Figure 3D) significantly decreased compared to the DNCB group. Dermal infiltration with mast cells and epidermal differentiation are hallmark features of changes in patients with AD [48], whereas ASE reversed these features. Thus, ASE could also play a role as a potent histological ameliorator of AD. The mRNA gene expressions of TNFα, IL-1β, IL-33, IL-25 and IL-4 were increased in the DNCB group (TNFα: 8.26 ± 1.99, IL-1β: 4.71 ± 1.03, IL-25: 3.41 ± 0.95, IL-33: 3.11 ± 0.23, IL-4: 13.4 ± 5.30 fold) compared to the CON group, whereas ASE significantly decreased gene expressions (TNFα: 3.18 ± 1.40, IL-1β: 1.16± 0.63, IL-25: 1.14 ± 0.24, IL-33: 1.33 ± 0.11, IL-4: 4.77 ± 1.49 fold) (Figure 3E). These results showed that application of ASE suppressed allergic immune responses. 

### 3.3. Isolation of Phytochemicals from ASE and Their Effects in RBL-2H3 Cells

Because ASE is an extract, it is composed of many compounds. Therefore, we investigated specific compounds within the extract that have ASE-like effects on atopic dermatitis. Acorns from the diverse *Quercus* species are reported to generally contain a variety of phenolics, such as gallic acid, ellagic acid, their derivatives and tannins [23]. Phytochemical analysis of ASE was confirmed by HPLC/MS analysis. Peaks were separated on the basis of their retention times and corresponded to fraction 1 (*m*/*z* 169 at t_R_ 4.2 min), fraction 2 (*m*/*z* 483, 613, 631, 635, 785, 787, 937, 939 at t_R_ 7.2~12.1 min), fraction 3 (*m*/*z* 301 at t_R_ 22.2 min) and fraction 4 (*m*/*z* 585 at t_R_ 30.8 min, Figure 4A). Based on these results, ASE (100 mg) was separated into fractions by recycling preparative HPLC. Recycle processing was repeated three times, and subsequently the last peak was classified into four fractions. The fractions were collected and evaporated under vacuum to yield fraction 1 (6.5 mg), fraction 2 (35.1 mg), fraction 3 (5.9 mg) and fraction 4 (4.0 mg). 

To examine the effects of ASE fractions, we investigated whether these fractions could regulate the pro-inflammatory cytokine IL-4 in RBL-2H3 cells. Fraction 1, 2 and 3 inhibited the mRNA expression of IL-4 compared to the PI-stimulated group in RBL-2H3 cells (Figure 4B). Among them, fraction 1 (10 µg/mL: 50.4 ± 4.3%, 30 µg/mL: 33.6 ± 5.5%) and fraction 3 (10 µg/mL: 36.9 ± 1.1%, 30 µg/mL: 19.9 ± 3.0%) significantly suppressed the mRNA expression of IL-4 lower than ASE (10 µg/mL: 66.6 ± 9.1%, 30 µg/mL: 45.3 ± 5.5%). β-hexosaminidase is secreted from the secretory granules of mast cells. When mast cells are immunologically activated, secretion of β-hexosaminidase is accompanied by histamine, so activation of β-hexosaminidase is used as a marker of mast cell degranulation [49]. To examine the effect of ASE, fraction 1 and fraction 3 on cell degranulation, we investigated β-hexosaminidase secretion by specific antigen–antibody reactions. Consistent with the inhibition of IL-4 mRNA expression, fractions 1 (10 µg/mL: 178.5 ± 20.6 and 30 µg/mL: 116.1 ± 17.9%,) and fraction 3 (10 µg/mL: 214.2 ± 27.6 and 30 µg/mL: 159.4 ± 7.6%) also significantly inhibited the release of β-hexosaminidase compared to IgE + DNP-BSA-stimulated cells (508.0 ± 9.8%) and ASE (10 µg/mL: 373.9 ± 28.2 and 30 µg/mL: 273.2 ± 23.3%) (Figure 4C). These results suggested that fractions 1 and 3 are the active fractions that inhibit both mRNA expression of IL-4 and the release of β-hexosaminidase in vitro. Th2 cells increase IgE synthesis by producing IL-4, IL-5, IL-6, IL-10 and IL-13 and inducing the differentiation of eosinophils and mast cells to induce type 1 hypersensitivity reactions. In our cell culture experiments, fractions 1 and 3 were determined to have anti-allergic effects through IL-4 inhibition. Furthermore, they also inhibited the β-hexosaminidase release by IgE stimulation. Thus, fractions 1 and 3 are expected to have an action similar to ASE in the AD-like skin lesion mouse model. These active fractions 1 and 3 from ASE were purified and identified as gallic acid and ellagic acid, respectively, through EI-MS, ^1^H- and ^13^C-NMR analysis (Figure 4D). 

IL-4 is a key driving factor for Th2 responses known to cause atopic dermatitis. When IL-4 expression and secretion are elevated in immune cells, the response of the sub-signals increases (e.g., STAT6 phosphorylation, etc.) and symptoms of atopic dermatitis appear. In addition, the efficacy of dupilumab, the drug that regulates IL-4, is known to be the best in the clinical trial. Previous papers have demonstrated that the extracts of natural products inhibiting IL-4 expression and some of the compounds isolated therefrom have excellent efficacy against atopic dermatitis in vivo [50,51] Therefore, compounds that inhibit IL-4 expression can be inferred to be able to ameliorate atopic dermatitis in vivo. Gallic acid and ellagic acid have previously been reported to have extensive anti-inflammatory activities, as revealed in our experiments. Gallic acid was shown to have anti-inflammatory effects by suppressing the hyperproduction of pro-inflammatory cytokines and the release of histamine in mast cells [52]. Gallic acid also has anti-inflammatory activities on acute food pad swelling in mice [53]. 

Ellagic acid has also been reported to attenuate IgE-mediated allergic responses including pro-inflammatory cytokine production on mast cells and in an asthma murine model [54,55]. These reports supported that gallic acid and ellagic acid, which are major phytochemicals of ASE, contribute to the in vivo anti-atopic effect of acorn shell extract in atopic dermatitis. Following of the above, we suggest that gallic acid and ellagic acid are active petrochemicals of ASE. Consequently, we expect that ASE and its major phytochemicals could be used as treatments for atopic dermatitis and inflammatory diseases.

## 4. Conclusions

We proved the improved effects of Acorn shell extract (ASE) on atopic dermatitis (AD) by oxazolone (OX)- or 2,4-dinitrochlorobenzene (DNCB)-induced AD-like lesion mouse models. In the OX-induced AD-like lesion Balb/c mice ear model, topical application of ASE improved the progression of AD-like lesions, including swelling. In addition, it also inhibited the augmentation of inflammatory cytokines, which lead to an allergic immune response. In the DNCB-induced AD-like lesion of SKH-1 hairless mice dorsal model, the application of ASE also improved of AD-like lesions, and it also significantly reversed skin barrier dysfunction and excessive production of serum IgE and Th2 cell expansion. Additionally, it has been demonstrated that the anti-atopic effects through the inhibition of inflammatory response and skin barrier dysfunction. Since ASE is an extract, we investigated specific active compounds from ASE. Among the various compounds of ASE, gallic acid and ellagic acid were found to down-regulate the inflammatory cytokine. Ultimately, we have demonstrated the potential of ASE and its major phytochemicals, including gallic acid and ellagic acid, as a remedy for atopic dermatitis and inflammatory diseases. 

## Figures and Tables

**Figure 1 biomolecules-10-00057-f001:**
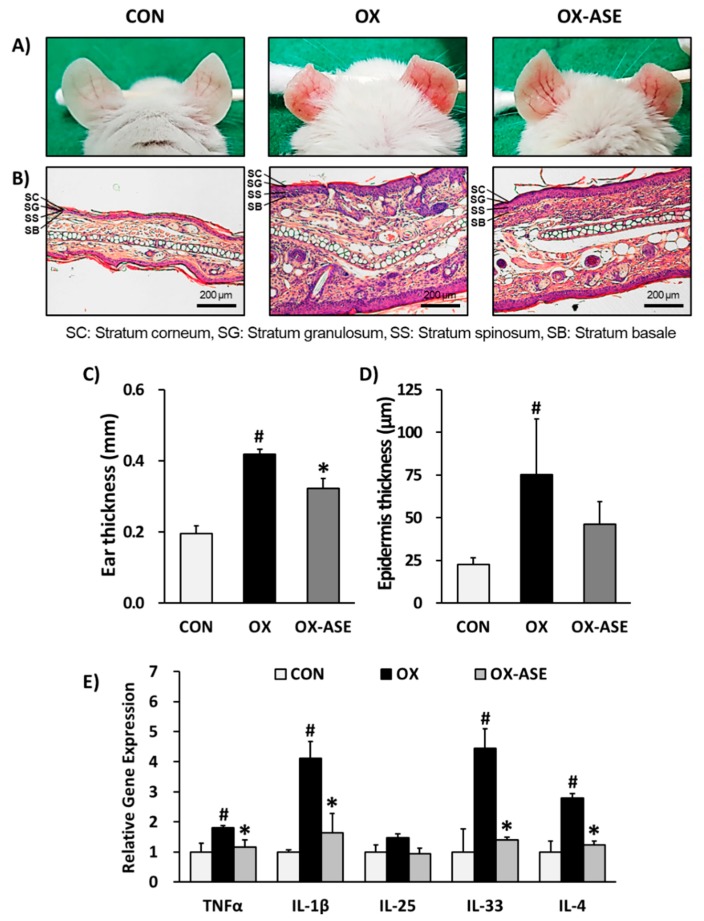
Acorn shell extract (ASE) improved atopic dermatitis (AD) in oxazolone-induced Balb/c mice. Clinical features of mouse ears (**A**). Hematoxylin and eosin (H&E) staining (200× magnification, **B**). Ear thickness (**C**). Epidermal thickness of ear (**D**). Relative gene expression of cytokines (TNFα, IL-1β, IL-25, IL-33, IL-4) (**E**). Data are presented as the mean ± standard deviation (SD), and were analyzed using a one-way analysis of variance (ANOVA) and a Tukey′s multiple comparisons post hoc analysis between the oxazolone-induced group (OX) and the other groups (CON, OX-ASE). # *p* < 0.05 significantly greater than control (CON). * *p* < 0.05 significantly lower than OX-ASE. CON, normal control group; OX: oxazolone-induced group, OX-ASE: oxazolone and acorn shell extract-treated group.

**Figure 2 biomolecules-10-00057-f002:**
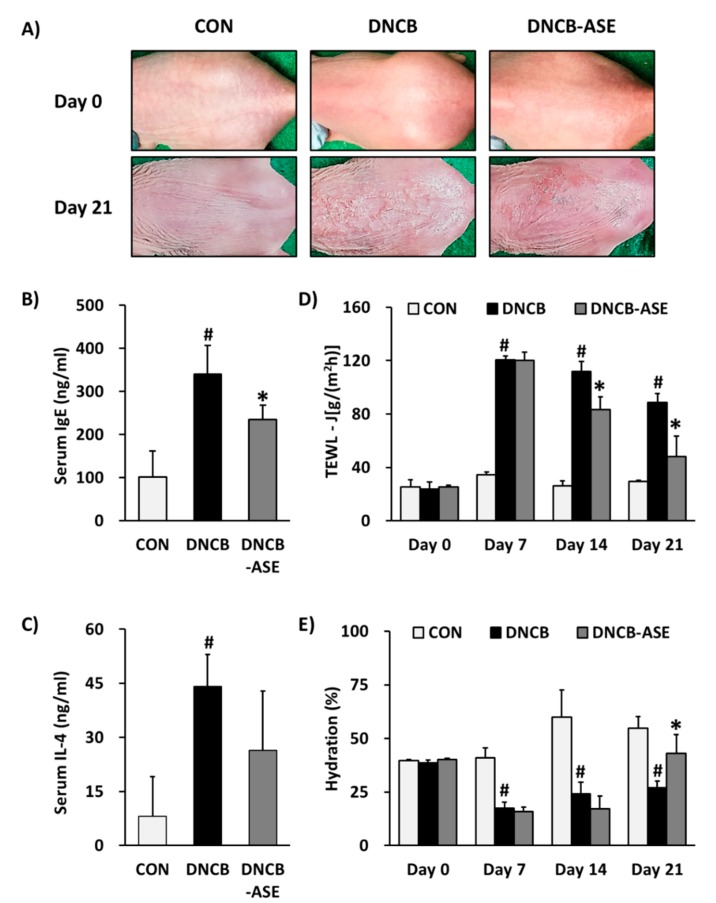
ASE-ameliorated pathological changes in the skin of DNCB-induced atopic hairless mice. Clinical features of AD-like skin lesions (**A**). Concentration of serum IgE (**B**). Concentration of serum IL-4 (**C**). Transepidermal water loss (TEWL) values (**D**). Level of skin hydration (**E**). Data are presented as the mean ± SD, and were analyzed using a one-way ANOVA and a Tukey′s multiple comparisons post hoc analysis between the DNCB-induced group (DNCB) and the other groups (CON, DNCB-ASE). # *p* < 0.05 significantly greater than CON. * *p* < 0.05 significantly lower than DNCB-ASE. CON, normal control group; DNCB: DNCB-induced group, DNCB-ASE: DNCB and acorn shell extract-treated group.

**Figure 3 biomolecules-10-00057-f003:**
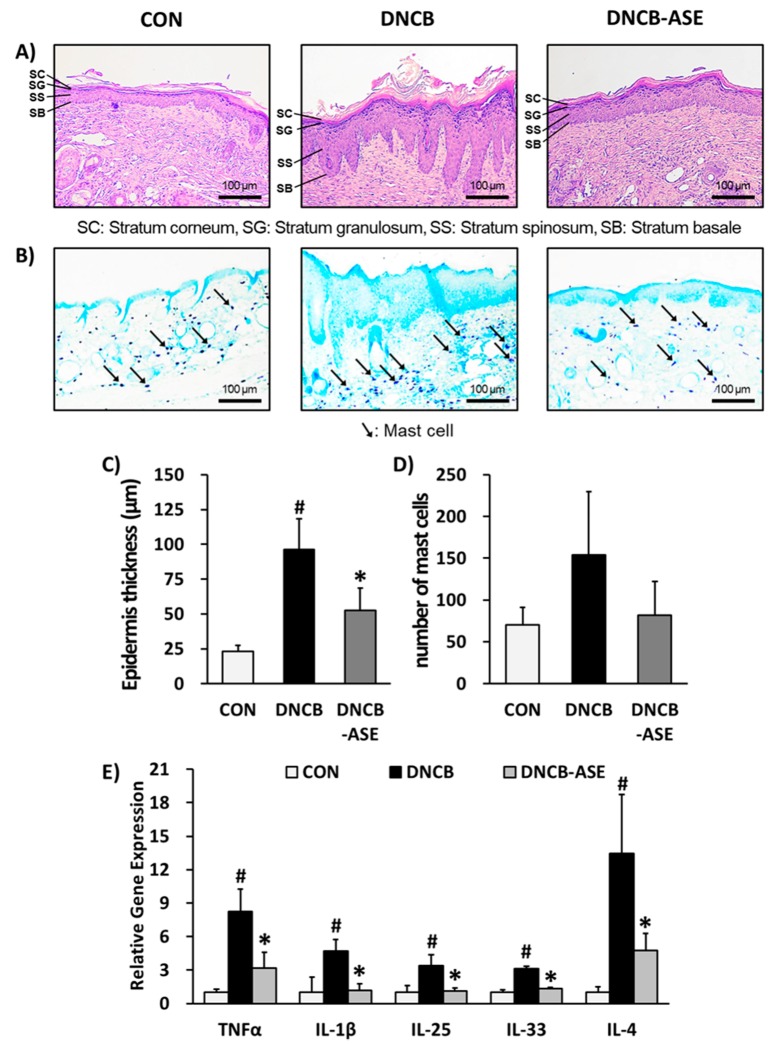
Histopathological effects of ASE in a DNCB-induced atopic hairless mouse model. Hematoxylin and eosin staining (200× magnification, (**A**). Toluidine blue staining (**B**). Epidermis thickness (**C**). Number of mast cells (**D**). Relative gene expression of cytokines (TNFα, IL-1β, IL-25, IL-33, IL-4) (**E**). Data are presented as the mean ± SD, and were analyzed using a one-way ANOVA and a Tukey’s multiple comparisons post hoc analysis between the DNCB-induced group (DNCB) and the other groups (CON, DNCB-ASE). # *p* < 0.05 significantly greater than CON. * *p* < 0.05 significantly lower than DNCB-ASE. CON, normal control group; DNCB: DNCB-induced group, DNCB-ASE: DNCB and acorn shell extract-treated group.

**Figure 4 biomolecules-10-00057-f004:**
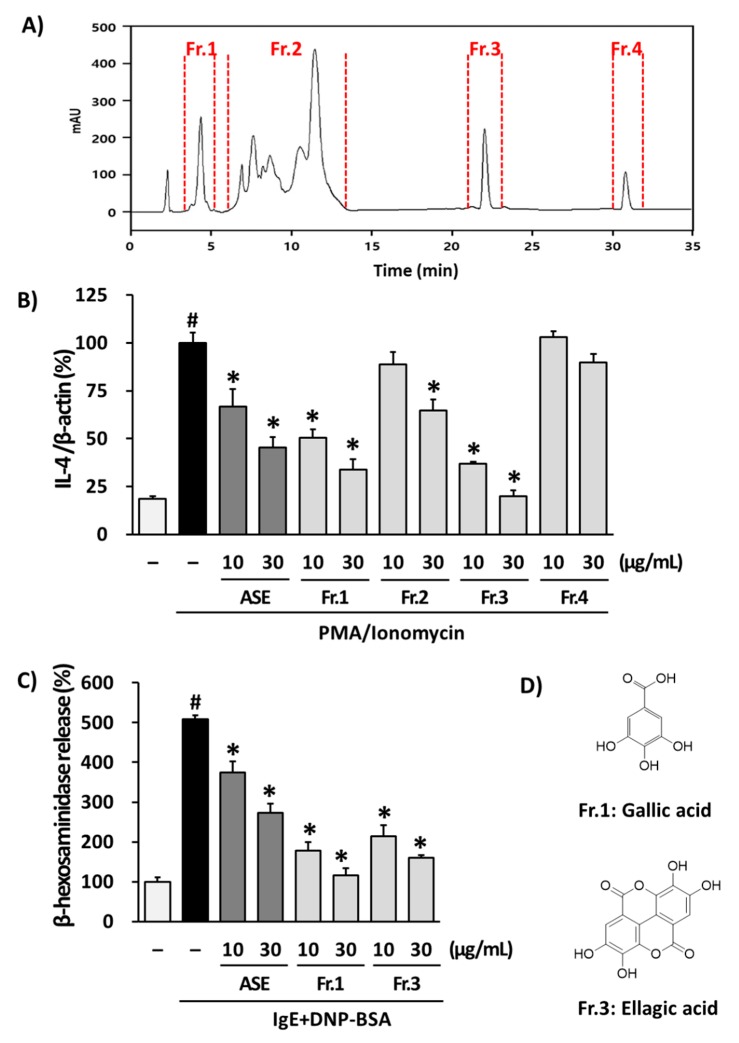
Anti-allergic effective acorn shell extract (ASE) fractions and active phytochemicals. HPLC chromatogram of acorn shell extract (ASE). Fr: fraction (**A**). Anti-allergic effects of ASE fractions in RBL-2H3 cells. IL-4 mRNA expression (**B**). β-hexosaminidase release (**C**). Chemical structures of active phytochemicals isolated from the acorn shell extract (ASE). Fr. 1: gallic acid; Fr. 3: ellagic acid (**D**). Data are presented as the mean ± SD, and were analyzed using a one-way ANOVA and a Tukey′s multiple comparisons post hoc analysis between the PMA/Ionomycin-treated group or IgE + DNP-BSA-stimulated group and the other groups. # *p* < 0.05 significantly greater than nontreated group. * *p* < 0.05 significantly lower than PMA/Ionomycin-treated group or IgE + DNP-BSA-stimulated group.

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
