# Peer review of "Anti-Atopic Effect of Acorn Shell Extract on Atopic Dermatitis-Like Lesions in Mice and Its Active Phytochemicals"

_biomolecules, 2019, doi:10.3390/biom10010057_

Round 1
Reviewer 1 Report
The topical treatment options of atopic dermatitis are limited. Therefore new substances are welcome.
However the authors should discuss the differences between mice and humans, which are important in such studies.
Are there other studies focussing on using acorn shell extract in humans? They mentioned TCM, are there any possibilities of toxic or allergic reactions?
Author Response
We appreciate reviewer’s valuable comment.
Point 1:
The topical treatment options of atopic dermatitis are limited. Therefore new substances are welcome. However the authors should discuss the differences between mice and humans, which are important in such studies.
Response1:
We added the discussion of differences between mice and humans in the Line 193-201.
“Haptens including oxazolone and DNCB, are small molecules that can effectively penetrate the skin barrier and epidermis of undamaged mouse skin. Proper exposure raises an acquired immune response, which causes a contact hypersensitivity reaction, such as allergic contact dermatitis. Repeated induction occur the type 1 and type 2 allergic contact dermatitis, creating a state like atopic dermatitis [35, 36]. However, the pathogenesis of allergic contact dermatitis and atopic dermatitis are not exactly the same. Therefore, it is difficult for these mouse models to clearly reappear atopic dermatitis. Although the mouse models are not a complete imitate of the overall clinical symptoms, it does provide a basic core of phenotypic expression [37, 38]. Thus, we tested the effects of acorn shell extract (ASE) using the mouse models that induces AD-like skin lesions with oxazolone and DNCB.”
Point 2:
Are there other studies focussing on using acorn shell extract in humans?
Response2:
In Iran, acorn shell has been used as a folk remedy for wound healing (Line 58). However, there are no other studies focusing on using acorn shell extract in humans.
Point 3:
They mentioned TCM, are there any possibilities of toxic or allergic reactions?
Response2:
There are many reports of allergic reactions after eating acorns. However, there are no reports of allergic reactions associated with skin exposure to acorns in people with nut allergy.
Reviewer 2 Report
Lee et al described a work about the anti-atopic effect of acorn shell extract on atopic dermatitis-like lesions in mice.
The experimental sections are well perfomed but an extensive editing of English was required, expecially in the Introduction part. For example :
"effective remedy for atopic dermatitis. Thus, the aim of this research is to propose acorn shells as a possible remedy for atopic dermatitis, since they are often abandoned and have anti-inflammatory effects". Remedy word is repeated two times between too very close sentences.
Figure 4: miss legend of x axis
Author Response
We thank reviewer for helpful comments.
Point 1:
Lee et al described a work about the anti-atopic effect of acorn shell extract on atopic dermatitis-like lesions in mice. The experimental sections are well perfomed but an extensive editing of English was required, expecially in the Introduction part. For example : "effective remedy for atopic dermatitis. Thus, the aim of this research is to propose acorn shells as a possible remedy for atopic dermatitis, since they are often abandoned and have anti-inflammatory effects". Remedy word is repeated two times between too very close sentences.
Response1:
We corrected the introduction section in the Line 38-66.
“Atopic dermatitis (AD) is a chronic inflammatory skin disease with symptoms including lesions from severe pruritus, epidermal hyperplasia, edema, erythema and erythematous plaque [1, 2]. The cause of AD has not yet been identified, but genetic factors, immune function imbalance and environmental factors are known to be involved [3, 4]. AD is estimated to occur in approximately 10-20% of the total population, but there is no clear curative treatment. Immunologically, activation of T lymphocytes, abnormalities in the cytokine system, decreases in cell mediated immunity, and increases in IgE have been reported to play an important role in the pathology of AD [5]. Atopic skin inflammation is regulated by diverse inflammatory cytokines. According to previous reports, the cytokines Interleukin 4 (IL-4), IL-5 and IL-13 are hyperproduced in AD patients compared with healthy patients, and IL-4 and Interferon-γ are clearly increased in acute AD [6, 7]. Currently, topical ointments and oral medications are used as AD therapies to suppress inflammation and itching [8]. Generally, steroids, antihistamines, immunosuppressants, and immunosuppressive calcineurin inhibitors are frequently used as therapeutics [8-10]. However, long term use of these agents causes side effects such as extreme atrophy of the skin, adrenal suppression and sensitivity to infection [9-13]. For these reasons, research is focused on identifying remedies for AD with few adverse effects. Therefore, we are conducting research focused on the development of new remedies originating from natural resources with low adverse drug reactions.
Acorns from up to 600 Quercus species are widely distributed worldwide and play an important role in food production and livestock husbandry [14-16]. Acorns have also been originally used in traditional Chinese medicine and folk medicine for diarrhea and diverse inflammatory disorders [17, 18]. In Iran, its shell have been used as a folk remedy for wound healing [19]. According to previous reports, various acorns have also been reported to have a variety of bioactivities, including free radical scavenging,[20] anti-bacterial activity,[21] anti-inflammatory activity [17] and antifungal activity [22]. However, the major components with these bioactivities are mainly secondary metabolites, which are found in shell more than in fruit [23]. Especially, the acorn shell has been reported to have angiogenesis activity in vitro [19]. Angiogenesis plays an important role in tissue regeneration for atopic diseases and the onset of allergic inflammation [24]. Thus, the aim of this research is to propose acorn shells as a possible remedy for atopic dermatitis, since they are often abandoned and have anti-inflammatory effects.”
Point 2:
Figure 4: miss legend of x axis
Response2:
We apologize for inadvertent error here. We added the legend of x axis in the figure 4.